# Playing Disability Rugby League with Charcot-Marie-Tooth Disease: A Case Study

**DOI:** 10.3390/sports11020021

**Published:** 2023-01-18

**Authors:** Luke Manny, Taylor Wileman, Che Fornusek, Daniel A. Hackett

**Affiliations:** Disciple of Exercise and Sport Science, Sydney School of Health Sciences, Faculty of Medicine and Health, The University of Sydney, Sydney 2006, Australia

**Keywords:** neuromuscular disorder, exercise, sports, physical function

## Abstract

Charcot-Marie-Tooth (CMT) disease is a common inherited neurological disorder that causes damage to peripheral nerves. Reports of CMT patients participating in team-based sports such as disability rugby league are scarce. The objective of this case report was to evaluate the benefits of participation in disability rugby league in a 50-year-old male with CMT. Leg muscle mass and strength was worse for the case subject compared to two age-matched CMT participants with an exercise history; however, evidence of greater function in the case subject was observed through better 6-min walk test performance. Performance in a series of sport specific tests was noticeably worse for the case subject compared to a fellow rugby league player (age matched) with cerebral palsy. Inferior in-game performance was observed for the case subject compared to his fellow rugby league player in terms of distance covered, top running speed, and intensity. However, the case subject may have assumed a different role when playing as evident by the different behaviours he displayed during the games (i.e., less player contacts, tackles, or touches, but more passes of the ball). This case study provides information concerning disability rugby league as an adjunctive mode of treatment for CMT populations.

## 1. Introduction

Charcot-Marie-Tooth disease (CMT) is a common inherited neurological disorder, affecting approximately 1 in 2500 people [1]. In Australia, this equates to approximately 10,144 people, although this figure is conservative given probable rates of underdiagnosis. CMT causes damage to the peripheral nerves which consist of both the motor and sensory nerves that connect the brain and spinal cord (the central nervous system) to the entire human body [2]. CMT is caused by mutations in genes that result in abnormal structure and function of either the peripheral nerve axon or the myelin sheath. There are approximately 80 genes identified in CMT disorders [3]. The most common form of this inherited demyelinating neuropathy is CMT type 1A, followed by X-linked CMT (CMTX), which is caused by duplication of the PMP22 gene [4]. The CMT variant influences the severity of muscle atrophy, weakness, and sensory problems [5]. The onset of CMT generally occurs over the initial two decades of life, first affecting the nerves at the farthest points of the body such as the hands and feet, and then progressing proximally [6]. Consequently, leading to limb muscle wasting, weakness, and sensory loss, with the lower limbs affected earlier and to a greater extent compared to the upper limbs. Foot and ankle muscle weakness is the single most debilitating problem for people with CMT making activities such as walking and jogging extremely difficult, with an increased risk of ankle sprains and tripping [7]. Disability for people with CMT relates to functional limitations and loss of independence, which negatively impacts their health-related quality of life (HRQoL) [8]. There is also evidence of significant sleep impairment and increased symptoms of depression in this clinical population [9]. Furthermore, the increased effort required to walk as well as the increased risk of falling and tripping may lead to low levels of physical activity, which is associated with the development of co-morbidities and all-cause mortality [10].

Sports participation is encouraged in rehabilitation programs to promote physical activity for disabled individuals [11]. Regular physical activity can improve overall fitness and function, preventing secondary health issues associated with sedentarism [12]. Organised sports can also have a positive influence on social and community participation, hence improving mental health [13]. For people with CMT, there is little evidence in the literature reporting on sports participation. A case study of a Paralympic swimmer with CMT reported positive transformations in physical and psychological parameters after five years of competitive activity compared to before initiating the sports activity [14]. However, to the authors’ knowledge, there has been no literature published on team sport participation in people with CMT. From an exercise perspective, the treatment for the symptoms and conditions associated with CMT has focused on aerobic and resistance training [15]. Both these forms of exercise have been shown to be effective for increasing strength and peak oxygen uptake, respectively, in this clinical population [16,17]. However, due to the prevalence of low-quality studies, the optimal dosage, type, and intensity of exercises remain unclear. For example, progressive resistance training has traditionally been prescribed at moderate intensities (≤70% one-repetition maximum—1RM) to mitigate injury risk and overwork weakness [18]. Thus, the safety and efficacy of a sport for people with CMT should be evaluated considering their specific symptoms that may influence performance and adherence. In addition, people with CMT have reported exercise intolerance and undue fatigue, which may present as major barriers to sport participation [19]. However, it should be noted that the reduced physical ability may not solely be related to the disease, but also due to physical deconditioning.

Rugby league is a popular sport in Australia, and a modified version of this sport was created to allow people with disabilities to participate. All eight physical paralympic movement impairment classes are eligible to play in the Physical Disability Rugby League (PDRL), however, those who have a visual or intellectual impairment are not eligible [20]. The game involves nine players per side playing on a smaller field over two periods of 20 min each. Of the nine players, a maximum of two able-bodied players can be on the field at one time. A maximum of two players wear red shorts to designate that they will play touch football rules (i.e., cannot be tackled). More details concerning the game can be accessed from the Playing Conditions, Physical Disability Rugby League manual [21]. The physically demanding nature of rugby league is preserved in the modified counterpart. Players are required to perform frequent bouts of high-intensity activity with short periods of low-intensity activity in the context of high-impact physical collisions [22]. Currently, there are no restrictions on worn devices to play the game. As such, players are allowed to wear any device that enables them to meet the demands of the game. In the case of players with CMT, ankle-foot-orthoses (AFO) would be used to correct foot-drop and ankle instability [23].

To date, there are no reports of CMT patients participating in team-based sports such as disability rugby league. Thus, the physical requirements for CMT patients to participate in disabled rugby league are currently unknown. This case study aims to present the physical characteristics and in-game physical performance of a 50-year-old male with CMT playing disability rugby league (CMT-RL). The fitness characteristics of CMT-RL were compared to two similarly aged, active CMT males. The performance of CMT-RL in a series of sport-specific fitness tests was compared to another player with cerebral palsy who participated in disability rugby league. This novel case report may provide helpful information to patients with CMT and their healthcare team to determine the suitability of participating in disability rugby league.

## 2. Materials and Methods

### 2.1. Case

CMT-RL is a 50-year-old male with the CMTX3 variant and was diagnosed at 8 years of age. The first symptom of onset began at the age of 3 years, with tiptoe walking, and by age 9 years the symptoms became clinically relevant as CMT-RL began to ambulate slower than his cohort. Hand involvement began at age 14 years with bilateral loss of thumb opposition and weakness in intrinsic hand muscles. CMT-RL’s past surgical history includes bilateral ankle arthrodesis (ankle fusion) surgeries, right hand reroute of the second digit tendon on the third finger to the thumb (improve thumb opposition), and right ankle debridement of osteophytes. CMT-RL has had a history of recurrent right ankle sprains, with a complete rupture of the anterior talofibular ligament (ATFL) at age 28 years, which was treated conservatively. CMT-RL participated in swimming and soccer as a child until 18 years of age. He transitioned into touch football and weight training, then from touch football to rugby league. He has been playing disability rugby league for 11 years. For 10 of those previous years, CMT-RL played contact; however, within the last year he began playing as a touch player. He currently uses AFOs for both lower limbs during gameplay, and he has been using AFOs or ankle braces since his ATFL rupture at age 28 years. CMT-RL does not actively engage in leisurely activity, only in purposeful exercise such as disability rugby league. Currently, CMT-RL runs 1–2 time a week for 20 min to improve fitness for playing disability rugby league.

### 2.2. Characteristics of Case Subject

For this case study, body composition, anthropometric measures, 6-min walk test (6MWT), handgrip strength, and 1RM for chest press and leg press was performed by CMT-RL and compared to two similarly aged, active CMT participants. The aged-matched CMT participants self-reported performing multiple sessions per week of resistance and aerobic exercise; hence, they were deemed to be physically active (CMT-Active). CMT-Active #1 presented with the CMTX variant, whereas CMT-Active #2′s variant was unknown. Body composition was assessed by fasting via a whole-body dual-energy X-ray absorptiometry scanner, (Lunar Prodigy, GE Medical Systems, Madison, WI, USA). The participants performed the 6MWT according to the American Thoracic Society guidelines [24]. Handgrip strength was assessed using a JAMAR handgrip dynamometer (Sammons Preston, Bolingbrook, IL, USA) following methods previously described [25]. Muscle strength (1RM) was assessed using the chest press and leg press, Keiser A420 pneumatic resistance training equipment (Keiser Sports Health Equipment, Inc., Fresno, CA, USA) [26]. Participants were provided with practice attempts for all physical performance tests to become familiarised prior to the actual testing. The data for the CMT-Active participants were collected during a previous study [26].

### 2.3. Sport-Specific Fitness Tests

The case subject performed a series of sport-specific fitness tests, active range of motion (ROM), and in-game performance measures. Another player with cerebral palsy (GMFCS-I) who participated in disability rugby league (CP-RL) completed the same testing. This subject was recruited due to having a similar amount of experience playing this sport and of a similar age (i.e., 52 years) to CMT-RL. All sport-specific fitness tests were performed on the same day, with the active range of motion measured prior to the other fitness tests.

The active range of motion of the right and left lower body limbs was measured according to the guidelines prescribed by Norkin and White [27]. Plantarflexion, dorsiflexion, knee flexion and extension, hip flexion, hip internal and external rotation, and hip abduction were measured twice using a universal goniometer, and the mean values were reported. The vertical jump test was performed as described by Reina et al. [28], using a jump mat (Fusion Sport, Coopers Plains, Australia). Each participant wore appropriate footwear and assistive devices as needed and were given three attempts, with the best measure taken. The isometric mid-thigh pull (lower body strength test) was performed using a C-Force Performance platform (Innovations, Western Australia, Australia) following the protocol prescribed by Comfort et al. [29]. Data for the isometric mid-thigh pull was collected using the software package (Ballistic Measurement System (BMS); Innervations, Perth, Australia). The variables recorded included peak force (N) and time to peak force (sec). Participants were familiarised with the vertical jump and mid-thigh pull prior to actual testing.

Maximal running speed was measured with a 20 m sprint, and change of direction speed (CODS) was assessed via the Illinois Agility Test. The timing gait system “Smart Speed” (Fusion Sport, Coopers Plains, Australia) was used for these tests. The gates were positioned at 0 m and 20 m for the sprint test and at the beginning and end of the course for the Illinois Agility Test. To familiarise subjects with the tests, a walk-through was performed twice before the actual trials. Each trial commenced with the subjects’ front foot 0.5 m behind the start line (and initial timing gate). The 20 m sprint commenced with a 2-point standing start, while the Illinois Agility Test commenced with the subjects lying prone on the ground. The fastest score from two trials for each respective test was used as the result.

### 2.4. In-Game Physical Performance Analysis

Two games of physical disability rugby league, which included an in-season and semi-final game, were video recorded and analysed (by a single observer). The game performance of CMT-RL and CP-RL was done retrospectively. The number of runs, tackles/touches, passes, errors, kicks, and missed tackles were recorded. A run was defined as the action of running with the ball before being touched/tackled, thus requiring the player to play the ball. An error was defined as an action by the attacking team that stopped play or stopped the attacking team from using the ball, referred to as a ‘knock-on’. For both games, CMT-RL and CP-RL wore “Apex Team Series” GPS tracking devices (STATSports, Newry, Northern Ireland). The distance covered and the top speed achieved were collected, and subsequently, the distance covered per minute was calculated retrospectively.

## 3. Results

### 3.1. Case Subject Versus Aged-Matched CMT Participants

The background and physical characteristics of CMT-RL and the age-matched CMT participants are presented in Table 1. The age of diagnosis was younger for CMT-RL (i.e., 8 years) compared to the CMT-Active subjects (i.e., ≥18 years). There were some noticeable differences in body composition between CMT-RL and the CMT-Active subjects. Leg lean mass was lower for CMT-RL compared to CMT-Active #1 (−31.5%) and CMT-Active #2 (−25.0%). Additionally, body fat percentage was highest for CMT-RL (32.0%) compared to the CMT-Active subjects (#1: 22.6% and #2: 28.8%). There were no obvious differences between the subjects for whole body and arm lean mass. CMT-RL covered a greater distance in the 6MWT compared to CMT-Active #1 (31.6%) and CMT-Active #2 (7.6%). Handgrip strength of CMT-RL was greater compared to CMT-Active #1 (~20%), but lower compared to CMT-Active #2 (~4 to 21%). Upper body strength was greater for CMT-RL compared to CMT-Active#1 (30.5%) but similar to CMT-Active #2. However, an obvious difference between CMT-RL and the CMT-Active subjects was identified for lower body strength. Leg press 1 RM was lower for CMT-RL compared to CMT-Active #1 (−61.3%) and CMT-Active #2 (−96.8%).

### 3.2. Sport-Specific Results—Case Subject Versus CP Rugby League Player

Results for the sport-specific tests and active ROM for CMT-RL and CP-RL are presented in Table 2. CMT-RL performed worse than CP-RL for the speed, agility, power, and lower body strength tests. CMT-RL was 29.4% and 21.1% slower than CP-RL in the 20 m sprint and Illinois agility tests, respectively. For the vertical jump test, CMT-RL jumped less than half the height of CP-RL (−115%). Additionally, CMT-RL compared to CP-RL produced less force in the isometric mid-thigh pull (−23.0%) as well as taking a longer time to reach peak force (33.3%). As for joint ROM, noticeable differences between CMT-RL and CP-RL were observed for the ankles. Plantarflexion of the ankles was ~50% less for CMT-RL compared to CP-RL. CP-RL displayed normal range, whereas CMT-RL could not actively dorsiflex either ankle. Hip internal rotation for CMT-RL was limited compared to CP-RL (i.e., ≥50% less ROM). There were no other discernible differences between CMT-RL and CP-RL for the ROM assessments.

### 3.3. In-Game Analysis—Case Subject Versus CP Rugby League Player

For both the in-season and semi-final games, CP-RL ran more than CMT-RL, whereas CMT-RL passed the ball more than CP-RL for these games (Figure 1). Additionally, players made more tackles/touches during the semi-final game compared to the in-season game.

CP-RL compared to CMT-RL covered more distance during both games (Figure 2a). Specifically, CP-RL covered 35% greater distance for the in-season game and 25.7% greater distance for the semi-final game. Higher top speed was achieved during both games by CP-RL (Figure 2b). The top speed for CP-RL compared to CMT-RL was 28.4% faster during the in-season game and 21.7% faster in the semi-final game. The distance covered per minute of gameplay is presented in Figure 2c. It should be noted that both players did not play the entire match. Distance covered was measured when the players were on the field playing. CP-RL covered more distance per minute than CMT-RL for both games (3.9% and 16.4% for the in-season and semi-final games, respectively).

## 4. Discussion

This novel case study presents a valuable insight into the physical characteristics and in-game performance of a 50-year-old male with CMT playing disability rugby league. There were some notable differences in the physical characteristics of the case subject compared to the age-matched active CMT participants. Leg muscle mass and strength were lower for CMT-RL, in addition to a greater body fat percentage. However, despite the appearance of impaired lower body function for CMT-RL, the case subject’s performance for the 6MWT was approximately >8% better than the active CMT participants; hence, indicating superior functional exercise capacity. The 6MWT distance achieved by CMT-RL was 203 metres better than the mean results reported in a recent study involving 168 people with CMT [30], and it was within the normative values for aged-matched healthy adults [31]. Since the case subject has a history of participating in rugby league, it is unknown whether involvement in this sport influenced his 6MWT performance. Rugby league involves intermittent bouts of high-intensity activity with short periods of low-intensity activities, which require speed, power, agility, cardiovascular fitness, and strength. Therefore, it is plausible that differences in performances for the 6MWT between CMT-RL and the active CMT participants may be due to sport-specific training effects.

Consistent with previously reported physical characteristics of people with CMT, the case subject displayed reduced ankle mobility, in both plantarflexion and dorsiflexion [32], with these values noticeably lower in comparison to CP-RL. Reduction in ankle mobility is known to negatively affect gait performance [33]. The only other noticeable difference between the case subject and CP-RL for joint mobility was hip internal rotation. Differences in hip kinematics in gait have been reported in children and adolescents with CMT. In a recent systematic review, a greater range of motion in flexion, abduction, and external rotation of the hip with a reduction in ankle ROM was reported [34]. The reduction in mobility for hip internal rotation in CMT-RL may be due to compensatory movements in the antagonist external rotator muscles to accommodate for the reduction in ankle mobility during his gait.

For all the sport-specific fitness tests, the performance of the case subject was inferior to CP-RL. The reduced ankle ROM for CMT-RL compared to CP-RL may have negatively affected their performance for the vertical jump, mid-thigh pull, and sprint tests. The battery of tests performed was chosen to give an indication of the specific physical requirements of rugby league. Therefore, it was not surprising that the match-play performance for CMT-RL was of a lower standard compared to CP-RL. During both the in-season and semi-final game, CP-RL compared to CMT-RL covered more distance, achieved a greater top speed, and performed at a greater intensity (distance covered/minute) than CMT-RL. However, performance for all these measures was greater for both players in the semi-final compared to the in-season game, suggesting increased physical demands during high stake games.

The in-game analysis revealed that the case subject was involved in fewer player contacts (i.e., tackles or touches) compared to CP-RL. However, he did pass the ball more than CP-RL during both games. Currently, the modified version of rugby league does not assign players to positions as seen in the traditional sport; therefore, it is unclear why these players displayed different behaviours during the games. One possible explanation could be that the players planned to adopt specific roles and responsibilities. Since CP-RL was involved in a lot more contact, his game activity is analogous to a forward position which has the primary role of running forward with the ball to gain as many meters as possible. In contrast, CMT-RL performed a lot more passes during the games, consistent with a playmaker role which involves the distribution of the ball and promotion of play for other players running the ball. It is also possible that the physical function of CMT-RL and CP-RL as well as their perceived physical abilities and related psychological factors (e.g., self-efficacy and confidence) influenced the activities performed by these players during the games.

For people with a disability, sports can facilitate the process of adapting to the disability [35]. The development of relationships among coaches and teammates can assist with improving communication skills and the ability to solve problems. The setting of sport-related goals can help provide a sense of personal strength and identity. Finally, participating in a sport can provide the person with a disability a sense of meaning and purpose through striving for mastery in sports and self-development. A longitudinal study conducted by Rasciute and Downward [36] in able-bodied participants aged ≥16 years found that sports participation had a statistically significant positive impact on health and happiness. Common psychological problems within the general population include anxiety, depression, stress, and low self-esteem. Therefore, it would be highly probable that sports participation would reduce the risk of these psychological problems [37]. From a sports performance perspective, improved self-confidence (e.g., skill mastery and physical abilities or motor performance) will generally lead to enhanced sporting performance [38]. The present case study provides an interesting insight for people with CMT considering participation in modified rugby league. CMT is a slowly progressive neuropathy that leads to both distal and proximal muscle weakness, with a distal-to-proximal progression [6,39]. Sensory impairments are also present which is characterised by nociceptive pain in the feet, lower limbs, and lumbar spine, and a decrease in proprioception leading to reduced coordination [5,17]. Due to impaired strength, power, coordination, and mobility, performing activities such as walking and running are difficult, with a heightened risk of tripping or twisting of the ankle [7]. However, for the case subject, despite the reduced speed, agility, power, and strength compared to CP-RL, he was able to accommodate these deficits by adopting a different playing role. Therefore, it appears that people with CMT can successfully participate in modified rugby league by spending more time on activities that involve less affected limbs (e.g., passing for the upper body), utilising assistive devices such as AFO’s, and having basic game knowledge of this sport. Additionally, despite the high-intensity nature of the sport, no adverse events or injuries were reported by CMT-RL.

To the authors best knowledge, there is no literature available on the feasibility for CMT populations to participate in team-based sports such as disability rugby league. Traditionally, aerobic exercises for people with CMT have been prescribed at submaximal loads, with the effects of high-intensity, supramaximal aerobic exercises limited [40]. This case study provides valuable insights into a 50-year-old CMT rugby league player performing high-intensity exercises as necessitated by the nature of the sport. Aligning with what has been reported in the recent case study of a paralympic CMT swimmer [14], participation in sport, either individually or team-based, may provide an adjunctive mode of treatment for CMT populations. However, further research on this topic is required to confirm the efficacy and safety of participating in modified rugby league for people with CMT.

Despite the novelty of this case study, there are many limitations to be acknowledged. A minimum standard or physical requirement needed to play the sport for people with CMT cannot be inferred. It is also difficult to gauge the effect of other factors such as experience in the sport, game knowledge, and skill of play on the in-game performance of the case subject. The sport-specific tests and active ROM tests were only measured on one occasion; therefore, it is possible that the results may not be reliable, thus representing the “true” abilities of CMT-RL and CP-RL. However, a strict testing protocol was followed, clear instructions were provided to the subjects, and multiple trials on the testing day were performed. Further, to ensure greater accuracy for the in-game physical requirements, a longitudinal study encompassing an entire playing season should be measured. There was also only one observer that analysed and annotated the game performance of CMT-RL and CP-RL using multiple views of the footage. However, since a second observer was not used to confirm the accuracy of the analysis, it is possible that some errors may have been made.

While CMT-RL and CMT-Active #1 had CMT variants CMTX and CMTX3, respectively, very similar presentation symptoms affecting their physical function are expected [5]. However, for CMT-Active #2, their CMT variant was unknown so it is possible that their performance could have been influenced by a different CMT type. It should be acknowledged that data from the CMT-Active participants were collected during a previous study, therefore it was not possible to replicate the tests completed for CP-RL. Additionally, body composition was not assessed for CP-RL which may have provided information concerning whether differences in lower body lean mass influenced vertical jump, mid-thigh pull, and sprint performance. Finally, in future studies, measures of fatigue and the impact on quality of life through playing disability rugby league should be assessed in people with CMT. This may provide further insight into the feasibility of people with CMT to play disability rugby league without any detrimental consequences to their health.

## 5. Conclusions

In conclusion, this case study provides information that could be helpful for people with CMT considering participation in disability rugby league. However, the report is specifically related to an individual with the CMTX variant. Although there is little research concerning the efficacy and safety for participating in this sport, promotion of physical activity is important for disabled individuals. Disability rugby league may provide an adjunctive mode of treatment for CMT populations. However, it is advised that a person with CMT is appropriately screened by an allied health professional and medical clearance is provided, if necessary, prior to participating in this sport.

## Figures and Tables

**Figure 1 sports-11-00021-f001:**
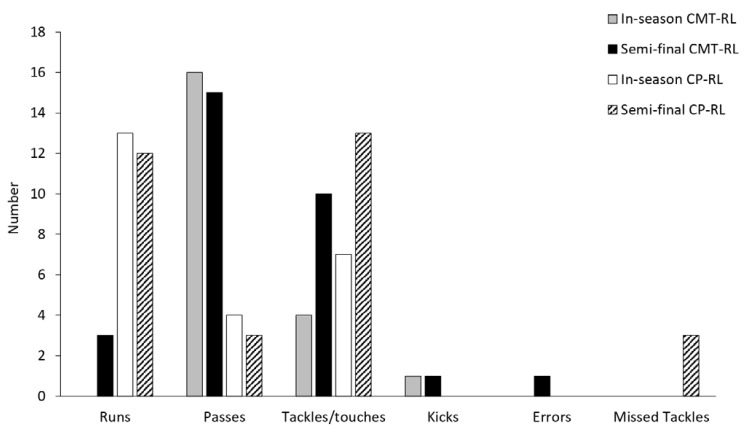
Performance of CMT-RL and CP-RL during two games in the 2022 NSWPDRLA season. CMT-RL: Charcot-Marie-Tooth disease rugby league player; CP-RL: cerebral palsy rugby league player. NSWPDRLA: NSW Physical Disability Rugby League Association.

**Figure 2 sports-11-00021-f002:**
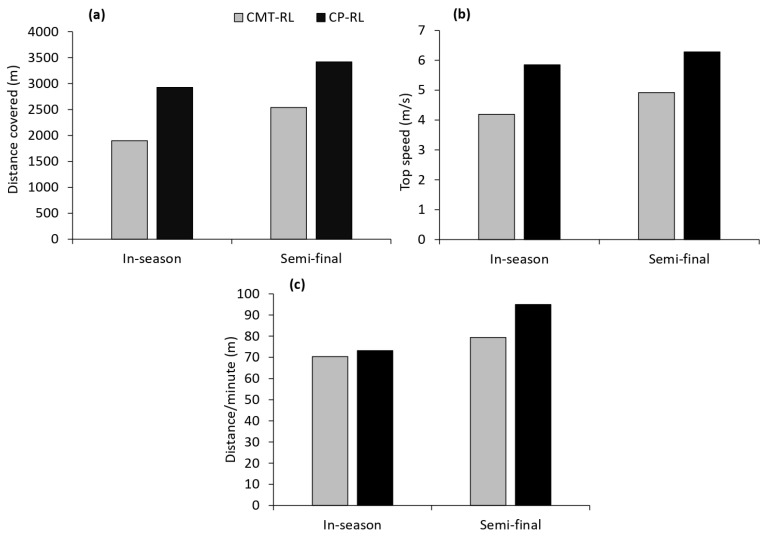
Performance characteristics of CMT-RL and CP-RL during two games in the 2022 NSWPDRLA season. (**a**) Distance covered during games; (**b**) top running speed during games; and (**c**) distance covered per minute of game play. CMT-RL: Charcot-Marie-Tooth disease rugby league player; CP-RL: cerebral palsy rugby league player. NSWPDRLA: NSW Physical Disability Rugby League Association.

**Table 1 sports-11-00021-t001:** Background and fitness characteristics of rugby league player with CMT and age-matched CMT active subjects.

	CMT-Active #1	CMT-Active #2	CMT-RL	Diff RL vs. Active #1 (%)	Diff RL vs. Active #2 (%)
Age (y)	57	54	50		
CMT type	CMTX	Unknown	CMTX3		
Age at diagnosis (y)	18	29	8		
Duration of diagnosis (y)	39	34	42		
BMI (kg/m^2^)	24.1	27.2	25.7	6.2	−5.8
Waist circumference (cm)	93.3	93.2	88.5	−5.4	−5.3
Lean body mass (kg) ^a^	54.4	53.2	54.1	−0.5	1.7
Arm lean mass (kg) ^a^	6.8	7.5	7.2	5.6	−4.2
Leg lean mass (kg) ^a^	16.3	15.5	12.4	−31.5	−25.0
% Body fat ^a^	22.6	28.8	32.0	29.4	10.0
6MWT (m)	408.3	551.6	597	31.6	7.6
Handgrip (kg)—right	15	23	19	21.1	−21.1
Handgrip (kg)—left	20	27	26	23.1	−3.8
1 RM Chest press (N)	452	640	650	30.5	1.5
1 RM Leg press (N)	2500	3050	1550	−61.3	−96.8

Data reported as means ± SD. BMI = body mass index; m = metres; kg = kilograms; cm = centimetres; % = percentage; CMT = Charcot-Marie-Tooth disease; RL = rugby league; y = years; 1 RM = one-repetition maximum; 6MWT = 6-min walk test. ^a^ Assessed via a dual-energy x-ray absorptiometry scanner (Lunar Prodigy, GE Medical Systems, Madison, WI) under fasted conditions. Diff CMT-RL versus Active # calculated via [(CMT-RL Minus Active #) divided by CMT-RL] multiplied by 100.

**Table 2 sports-11-00021-t002:** Comparison between CP and CMT rugby players of similar age for sport-specific fitness tests.

	CP-RL	CMT-RL	Diff CMT-RL vs. CP-RL (%)
20 m sprint (sec)	3.6	5.1	29.4
Illinois agility (sec)	20.9	26.5	21.1
Vertical jump (cm)	23	10.7	−115
Isometric mid-thigh pull (N)	2358.8	1917.2	−23.0
Time to peak force (sec)	1.6	2.4	33.3
Joint range of motion (degrees)			
Plantar flexion—right Plantar flexion—left	39 38.5	25 25	−56.0 −54.0
Dorsi flexion—right Dorsi flexion—left	20 16.5	−15 ^a^ −14 ^a^	−233.0 −218.0
Knee flexion—right Knee flexion—left	140 131.5	135.5 137.5	−3.3 4.4
Knee extension—right Knee extension—left	−3 ^a^ 0.5	1 4	400 87.5
Hip flexion—right Hip flexion—left	114 103	109 105	−4.6 1.9
Hip internal rotation—right Hip internal rotation—left	43.5 42	26.5 28	−64.2 −50.0
Hip external rotation—right Hip external rotation—left	40.5 36	48.5 36.5	16.5 1.4
Hip abduction—right Hip abduction—left	25.5 34.5	33 33.5	22.7 −3.0

CMT = Charcot-Marie-Tooth disease; CP = cerebral palsy; RL = rugby league. Diff = difference. Diff CMT-RL versus CP-RL calculated via [(CMT-RL Minus CP-RL) divided by CMT-RL] multiplied by 100. ^a^ neutral position (i.e., 0 degrees) was not achieved or surpassed explaining the negative numbers. As an example, dorsiflexion (right) for CMT-RL was −15 degrees from achieving a neutral position compared to 20 degrees from achieving a neutral position for CP-RL.

## Data Availability

The datasets generated during the current study are available from the corresponding author on reasonable request.

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
