# Peer review of "Playing Disability Rugby League with Charcot-Marie-Tooth Disease: A Case Study"

_sports, 2023, doi:10.3390/sports11020021_

Round 1

Reviewer 1 Report

This paper is a case report, a 50-year-old-male with Charcot-Marie-Tooth (CMT) disease (CMTX3 variant) with a history of participation in disability rugby league. The scientific article is innovative and the accuracy is adequate. Currently, there are few study of CMT patients participating in team-based sports. Title is appropriate and the introduction clearly sum up the aim of the study.

The data reported in Materials and Methods are complete (Case, Characteristics of Case Subject, Sport Specific Fitness Tests, In-Game Physical Performance Analysis). The results (Case Subject versus Aged-Matched CMT Participants, Sport Specific Results – Case Subject versus CP rugby league player, In-Game Analysis – Case Subject versus CP rugby league player) are reported clearly and concisely.

Adequate is the discussion that sum up the main results, critically analyse the methods used, compare the results obtained and discuss the implications of the results. The conclusions are justified by the data.

Tables 1 (Background and fitness characteristics of rugby league player with CMT and age matched CMT active subjects), Tables 2 (Comparison between CP and CMT rugby players of similar age for sport specific fitness tests) Figure 1 (Performance of CMT-RL and CP-RL during two games in the 2022 NSWPDRLA season) and Figure 2 (Performance characteristics of CMT-RL and CP-RL during two games in the 2022 NSWPDRLA season) sum up research content correctly.

Areas of criticism to consider:

·         the discussion (section number 4), if possible, should be expanded with the psychological aspects of sport and the correlation with possible changes on performance

Author Response

We would like to express our gratitude to you for reviewing our manuscript and providing feedback.

Comment: The discussion (section number 4), if possible, should be expanded with the psychological aspects of sport and the correlation with possible changes on performance.

Response: Thank you for providing this suggestion. We have added a paragraph to address the topic of psychological aspects related to sports for people involved in disability sports (please see below).

“For people with a disability, sports can facilitate the process of adapting to the disability [35]. The development of relationships among coaches and teammates can assist with improving communication skills and the ability to solve problems. The setting of sport-related goals can help provide a sense of personal strength and identity. Finally, participating in a sport can provide the person with a disability a sense of meaning and purpose through striving for mastery in sports and self-development. A longitudinal study conducted by Rasciute and Downward [36] in able-bodied participants aged ≥16 years found that sports participation had a statistically significant positive impact on health and happiness. Common psychological problems within the general population include anxiety, depression, stress, and low self-esteem. Therefore, it would be highly probable that sports participation would reduce the risk of these psychological problems [37]. From a sports performance perspective, improved self-confidence (e.g., skill mastery and physical abilities or motor performance) will generally lead to enhanced sporting performance [38].”

Reviewer 2 Report

The paper Playing Disability Rugby League with Charcot-Marie-Tooth 2 Disease: A Case Study by Manny et al discusses the potential beneficial effects of disability rugby in a 50-year-old male subject with Charcot-Marie-Tooth 2 disease. Overall, the paper is interesting and adds an important contribution to the current literature. However, some important revisions should be addressed.

Abstract

This section should contain a brief background on CMT and should highlight the importance of the study and what important new information this adds to current knowledge in this field. In addition, it is important to add the purpose of the work in this section. I suggest that this section be rewritten.

Introduction

This section is clear and provides the appropriate context. However, since two different types of controls are used in the study, two physically active CMT subjects and one subject with cerebral palsy who participated in the disability rugby league, it is necessary to clarify the study design by specifying which comparative analyses that were performed and the role of all subjects enrolled in the study.

Materials and Methods - Case Subject Characteristics.

The CMT variant of physically active CMT subjects needs to be specified in this section.

Results

The variant of the active CMT subjects was different (or even unknown) from that of the case subject. Could this affect the baseline characteristics and results of the study?

Conclusions

If the study contains limitations, they should be included.

Author Response

We would like to express our gratitude to you for reviewing our manuscript and providing feedback.

Comment 1: (Abstract) This section should contain a brief background on CMT and should highlight the importance of the study and what important new information this adds to current knowledge in this field. In addition, it is important to add the purpose of the work in this section. I suggest that this section be rewritten.

Response: The information suggested has now been included. Please keep in mind that the journal has a 200-word limit for the Abstract. The new information added is presented below.

“Charcot-Marie-Tooth (CMT) disease is a common inherited neurological disorder that causes damage to peripheral nerves. Reports of CMT patients participating in team-based sports such as disability rugby league are scarce. This case study aims to present the physical characteristics and in-game physical performance of a 50-year-old male with CMT playing disability rugby league (CMT-RL).”

“This case study provides information concerning disability rugby league being an adjunctive mode of treatment for CMT populations.”

Comment 2: (Introduction) This section is clear and provides the appropriate context. However, since two different types of controls are used in the study, two physically active CMT subjects and one subject with cerebral palsy who participated in the disability rugby league, it is necessary to clarify the study design by specifying which comparative analyses that were performed and the role of all subjects enrolled in the study.

Response: We agree with your suggestion and have included the relevant information in the Introduction. Please see below.

“Fitness characteristics of CMT-RL were compared to two similarly aged, active CMT males. Performance of CMT-RL in a series of sport-specific fitness tests was compared to another player with cerebral palsy who participated in disability rugby league.”

Comment 3: (Materials and Methods - Case Subject Characteristics) The CMT variant of physically active CMT subjects needs to be specified in this section.

Response: We have moved the CMT variant information from the Results to the Materials and Methods section. Please see below.

“CMT-Active #1 presented with the CMTX variant, whereas CMT-Active #2’s variant was unknown.”

Comment 4: (Results) The variant of the active CMT subjects was different (or even unknown) from that of the case subject. Could this affect the baseline characteristics and results of the study?

Response: CMT-active #1 with CMTX would likely have very similar presentation symptoms affecting physical function as CMT-RL. However, for the CMT-Active #2 with the unknown variant, it is possible that their performance could be influenced by a different CMT type. This has been clarified in the Discussion – see below.

“While CMT-RL and CMT-Active #1 had CMT variants CMTX and CMTX3, very similar presentation symptoms affecting their physical function are expected [5]. However, for CMT-Active #2, their CMT variant was unknown so it is possible that their performance could have been influenced by a different CMT type.”

Comment 5: (Conclusions) If the study contains limitations, they should be included.

Response: We have added this information to the conclusion (see below).

“However, the report is specifically related to an individual with the CMTX variant.”

We have also expanded upon the limitations section.

Reviewer 3 Report

Thank you for the invitation to review this case study on a gentleman with Charcot-Marie-Tooth disease participating in Disability Rugby League. I found the study interesting and believe there is merit in data collected in this and similar populations regarding exercise participation and performance. I have a few suggestions which I believe could improve the case study.

Abstract

I would recommend summarising the aim of the study at the beginning of the abstract, to provide the reader with clear rationale for this case study. Lifting a sentence from the final paragraph of the introduction would be appropriate.

If word limit becomes an issue after including the above, I suggest the final sentence of the abstract may not be needed, as this is suggested within the manuscript.

Introduction

The introduction is useful and comprehensive for a reader with little knowledge of CMT, with the prevalence, symptoms and implications of the condition well described. However, it would be useful to add some context that CMT arises due to genetic mutations in a number of genes, and that several variants exist.

Currently the concept of different CMT variants only becomes clear later in the manuscript. I would suggest 2-3 sentences within the introduction to make clear that a number of genes (with some examples or citations) and mutations are involved in CMT (and how many), which are the most common or prevalent CMT subtypes, and whether different subtypes can present with a varying range/severity of impact on mobility etc. These are important points for the context of this case study, particularly around the ability to generalise the findings to individuals who have different CMT variants.

Materials & Methods

Lines 115-119: Were participants familiarised with hand grip, 1RM, vertical jump and mid-thigh pull tests, or did participants already have experience? It is mentioned that familiarisation was performed for running tests, but not mentioned for these measures. If not, is it possible that a learning effect could be present?

Lines 117-119: What protocol was used for 1RM testing? Please add a citation to support the protocol selected.

Lines 148-149: How many researchers analysed/annotated the game performance CMT-RL and CP-RL? Was this a single observer, or were any verification steps taken? (i.e a second observer, with comparison of agreement between the two, single observer viewing the footage multiple times etc). Depending on how this analysis was conducted, it may be appropriate to mention as a limitation.

Results

Table 1/Lines 161-162 – CMT-Active has CMT type CMTX3 and CMT-RL has CMTX. Are these exactly the same, or would differences (physical function, etc) be expected between each? Please clarify in the discussion.

Line 168-169 – I suggest re-wording the sentence to explain that CMT-RL covered a greater distance in the 6-min walk test, rather than simply that performance was greater.

Line 172 – at the end of the sentence it may be more appropriate to say that performance was “similar” to CMT-Active #2, rather than “like”.

Table 2 – Rate of force development is mentioned, but it is not clear where, when and how this was measured. Please add to methodology and results whether this was lower/upper body, and which test/equipment was used to capture/calculate this metric. Also – are the measurement units (seconds) correct? 

Discussion

Is it possible to specify/discuss why different measures were collected for comparisons between CMT-RL and CMT-Active (#1 and #2) versus those between CMT-RL and CP-RL? It may have been of interest to know the range of motion measures for the active CMT subjects, as this could potentially have affected the differences in some results (e.g leg press and 6-minute walk).

Similarly, the characteristics of CP-RL would also be of interest if available, as some characteristics could have influenced sport-specific test performance (e.g might differences in lower body lean mass impact vertical jump/sprint/mid-thigh pull performance?. Both of the above could be discussed as a limitation if data are not available.

Line 243-244 – could reduced ankle ROM have negatively impacted the performance of CMT-RL versus CP-RL in any other tests? One would suggest that acceleration (affecting 20m sprint) and vertical jump performance would be impaired by the effects of reduced ankle ROM on lower-body contractile activity.

Conclusions

Conclusions are appropriate and summarise the results of this case study well. As mentioned earlier, if there is known to be a notable difference in functionality and/or exercise ability between different CMT variants, it may be appropriate to make clear that the current case study reports specifically on an individual with “CMTX” variant.

Author Response

We would like to express our gratitude to you for reviewing our manuscript and providing feedback.

Comment 1: (Abstract) I would recommend summarising the aim of the study at the beginning of the abstract, to provide the reader with clear rationale for this case study.

Response: The information suggested has now been included. The new information added is presented below.

“Charcot-Marie-Tooth (CMT) disease is a common inherited neurological disorder that causes damage to peripheral nerves. Reports of CMT patients participating in team-based sports such as disability rugby league are scarce. This case study aims to present the physical characteristics and in-game physical performance of a 50-year-old male with CMT playing disability rugby league (CMT-RL).”

Comment 2: (Introduction) It would be useful to add some context that CMT arises due to genetic mutations in a number of genes and that several variants exist. I would suggest 2-3 sentences within the introduction to make clear that a number of genes (with some examples or citations) and mutations are involved in CMT (and how many), which are the most common or prevalent CMT subtypes, and whether different subtypes can present with a varying range/severity of impact on mobility etc.

Response: Thank you for this suggestion. We have added the following to the Introduction.

“CMT is caused by mutations in genes that result in abnormal structure and function of either the peripheral nerve axon or the myelin sheath. There are approximately 80 genes identified in CMT disorders [3]. The most common form of this inherited demyelinating neuropathy is CMT type 1A, followed by X-linked CMT (CMTX), which is caused by duplication of the PMP22 gene [4]. The CMT variant influences the severity of muscle atrophy, weakness, and sensory problems [5].”

Comment 3: (Materials & Methods) Lines 115-119: Were participants familiarised with hand grip, 1RM, vertical jump and mid-thigh pull tests, or did participants already have experience? It is mentioned that familiarisation was performed for running tests, but not mentioned for these measures. If not, is it possible that a learning effect could be present?

Response: Thank you for picking up this error. Participants were familiarised with all testing and the sentences below have been added to this section to make this clear to the reader.

“Participants were provided with practice attempts for all physical performance tests to become familiarised prior to the actual testing.”

“Participants were familiarised with the vertical jump and mid-thigh pull prior to actual testing.”

Comment 4: (Materials & Methods) Lines 117-119: What protocol was used for 1RM testing? Please add a citation to support the protocol selected.

Response: The following protocol used to conduct the 1RM testing has now been provided (see below).

“Roberts-Clarke, D.; Fornusek, C.; Saigal, N.; Halaki, M.; Burns, J.; Nicholson, G.; Fiatarone Singh, M.; Hackett, D. Relationship between physical performance and quality of life in Charcot-Marie-Tooth disease: a pilot study. J Peripher Nerv Syst 2016, 21, 357-364.”

Comment 5: (Materials & Methods) Lines 148-149: How many researchers analysed/annotated the game performance CMT-RL and CP-RL? Was this a single observer, or were any verification steps taken? (i.e a second observer, with comparison of agreement between the two, single observer viewing the footage multiple times etc). Depending on how this analysis was conducted, it may be appropriate to mention as a limitation.

Response: A single observer analysed/annotated the game performance CMT-RL and CP-RL, with footage viewed multiple times. However, it would have been ideal if a second observer was used to verify the results. Therefore, we have included this point as a limitation of the study (see below).

“There was also only one observer that analysed and annotated the game performance of CMT-RL and CP-RL using multiple views of the footage. However, since a second observer was not used to confirm the accuracy of the analysis it is possible that some errors may have been made.”

 Comment 6: (Results) Table 1/Lines 161-162 – CMT-Active has CMT type CMTX3 and CMT-RL has CMTX. Are these exactly the same, or would differences (physical function, etc) be expected between each? Please clarify in the discussion.

Response: CMT-active #1 with CMTX would likely have very similar presentation symptoms affecting physical function as CMT-RL. However, for the CMT-Active #2 with the unknown variant, it is possible that their performance could be influenced by a different CMT type. This has been clarified in the Discussion – see below.

“While CMT-RL and CMT-Active #1 had CMT variants CMTX and CMTX3, very similar presentation symptoms affecting their physical function are expected [5]. However, for CMT-Active #2 their CMT variant was unknown so it is possible that their performance could have been influenced by a different CMT type.”

Comment 7: (Results) Line 168-169 – I suggest re-wording the sentence to explain that CMT-RL covered a greater distance in the 6-min walk test, rather than simply that performance was greater.

Response: We have made the suggested amendment (please see below).

“CMT-RL covered a greater distance in the 6-minute walk test compared to CMT-Active #1 (31.6%) and CMT-Active #2 (7.6%).”

Comment 8: (Results) Line 172 – at the end of the sentence it may be more appropriate to say that performance was “similar” to CMT-Active #2, rather than “like”.

Response: The suggestion has been incorporated (see below).

“……similar to CMT-Active #2.”

Comment 9: Table 2 – Rate of force development is mentioned, but it is not clear where, when and how this was measured. Please add to methodology and results whether this was lower/upper body, and which test/equipment was used to capture/calculate this metric. Also – are the measurement units (seconds) correct? 

Response: Information has been added to the Materials and Methods section to address your comment. The units are correct for ‘time to peak force’. There was an error with initially naming this variable 'rate of force development'. Please see the changes made below.

“The isometric mid-thigh pull (lower body strength test) was performed using a C-Force Performance platform (Innovations, Western Australia, Australia) following the protocol prescribed by Comfort et al [29]. Data for the isometric mid-thigh pull was collected using the software package (Ballistic Measurement System (BMS); Innervations, Perth, Australia). The variables recorded included peak force (N) and time to peak force (sec). Participants were familiarised with the vertical jump and mid-thigh pull prior to actual testing.”

Comment 10: (Discussion) Is it possible to specify/discuss why different measures were collected for comparisons between CMT-RL and CMT-Active (#1 and #2) versus those between CMT-RL and CP-RL? It may have been of interest to know the range of motion measures for the active CMT subjects, as this could potentially have affected the differences in some results (e.g leg press and 6-minute walk).

Response: The results from the CMT-Active participants were collected during a previous study. The following information was included in the Materials and Methods section:

“The data for the CMT-Active participants were collected during a previous study [26].”

Limitations – Discussion:

“It should be acknowledged that data from the CMT-Active participants were collected during a previous study, therefore it was not possible to replicate the tests completed for CP-RL. Additionally, body composition was not assessed for CP-RL which may have provided information concerning whether differences in lower body lean mass influenced vertical jump, mid-thigh pull, and sprint performance.”

Comment 11: (Discussion) Similarly, the characteristics of CP-RL would also be of interest if available, as some characteristics could have influenced sport-specific test performance (e.g might differences in lower body lean mass impact vertical jump/sprint/mid-thigh pull performance? Both of the above could be discussed as a limitation if data are not available.

Response: We have added this as a limitation (see below).

“It should be acknowledged that data from the CMT-Active participants were collected during a previous study, therefore it was not possible to replicate the tests completed for CP-RL. Additionally, body composition was not assessed for CP-RL which may have provided information concerning whether differences in lower body lean mass influenced vertical jump, mid-thigh pull, and sprint performance.”

Comment 12: Line 243-244 – could reduced ankle ROM have negatively impacted the performance of CMT-RL versus CP-RL in any other tests? One would suggest that acceleration (affecting 20m sprint) and vertical jump performance would be impaired by the effects of reduced ankle ROM on lower-body contractile activity.

Response: We agree with your point raised and have included the following sentence to the Discussion.

“The reduced ankle ROM for CMT-RL compared to CP-RL may have negatively affected their performance for the vertical jump, mid-thigh pull, and sprint tests.”

Comment 13: (Conclusion) Conclusions are appropriate and summarise the results of this case study well. As mentioned earlier, if there is known to be a notable difference in functionality and/or exercise ability between different CMT variants, it may be appropriate to make clear that the current case study reports specifically on an individual with “CMTX” variant.

Response: We have added this information to the conclusion (see below).

“However, the report is specifically related to an individual with the CMTX variant.”

Round 2

Reviewer 2 Report

Overall, the authors have fulfilled the requirements. 

However, in the abstract section, the purpose of the paper is not clear. Why is it important to know the physical characteristics and physical performance of a rugby player with CMT?  If the objective of the paper is to evaluate the benefits of rugby in CMT it should be reported (within 200 characters).

Author Response

Comment: In the abstract section, the purpose of the paper is not clear. Why is it important to know the physical characteristics and physical performance of a rugby player with CMT?  If the objective of the paper is to evaluate the benefits of rugby in CMT it should be reported (within 200 characters).

Response: We agree with this suggestion and have amended it accordingly (see below).

"The objective of this case report was to evaluate the benefits of participation in disability rugby league in a 50-year-old male with CMT."